

# Evaluation of Extreme Sea-Levels and Flood Return Period using Tidal Day Maxima at Coastal Locations in the United Kingdom

Stephen E. Taylor[1]

[1]Geomatix Ltd. UK

*Correspondence to*: Stephen E. Taylor (set@geomatix.net)

**Abstract.** Tidal storm surges can result in significant damage and inundation if sea defences are insufficiently robust. Coastal planners need to know the risk of flooding so that sea defences and coastal developments can be specified and sited appropriately. Since Gumbel's original work on extreme value statistics, several modifications and new methods have been proposed for evaluating the risk of tidal inundation, with the Skew Surge Joint Probability Method (SSJPM) recently gaining

popularity. However, SSJPM is complex, often requiring manual intervention, and is difficult to automate. Guided by the search for a method specifically applicable to tides that is amenable to automation, this paper proposes several modifications to Gumbel's original approach. The novel technique is termed TMAX since its initial time unit is one tidal day, rather than the usual annual maxima (AMAX). Compared to AMAX, the TMAX method offers more efficient use of extreme data events, provides reduced variance in design height, and more efficiently handles missing data. The results of TMAX are compared

with those of a recent study using the SSJPM method at 35 United Kingdom identical coastal locations, showing broad agreement. This new approach provides a robust mechanism for extreme tide analysis and better informs strategies for coastal management and resilience.

## 1 Introduction

Coastal planners and developers require accurate estimates of coastal flood risk to design and site sea defences, coastal

buildings, harbours, nuclear power stations, and other critical infrastructure. Coastal floods generally occur when three factors converge; a significant storm surge, a spring tide and the time of high tide occurs at or near the peak of the storm surge. It is predicted the frequency of such events will double by 2050 due to rising sea levels (Vitousek et al. 2017). This challenge is especially critical for low-lying coastal nations, including parts of the United Kingdom (UK), (Williams 2016).

The required height of sea defences can vary quite rapidly along the coastline, as the coastal seabed topography can

magnify or diminish tidal and surge effects. Coastal tide gauges serve as vital data sources for estimating flood probabilities and return periods, providing the local records necessary for deriving flood risk parameters. Tide gauge data is often used for tidal harmonic analysis, and the establishment of tidal levels such as the Highest Astronomical Tide, (HAT). However, extreme sea-level analysis can leverage the same tide-gauge data to accurately predict the variation of flood return period with height. It is important to distinguish between HAT and extreme flood height. HAT inherently assumes that tidal height noise averages





out, whereas flood defence heights, take into account the effect of noise. Several statistical methods have been developed to make this assessment, which are now briefly reviewed.

    The Gumbel Type I AMAX uses only the largest annual extreme events for its extreme value analysis; with significant events that rank second or lower not being utilised. To address this apparent inefficient use of available data, several alternative techniques have been developed. One such technique is the "r-largest" method (Smith 1986; Tawn 1988), which involves using

a fixed number, r, of the largest annual maxima for analysis. Another approach is the "Peaks over Threshold" (POT) method described by Coles (2001), which utilizes maxima that exceed a certain threshold value to create a new probability distribution. A different approach by Pugh and Vassie (1978) splits the total sea level into two components: the deterministic astronomical tide component and a random surge component. Each component is converted into separate probability distributions which are then combined by convolution to produce a joint probability distribution function (PDF). The method is known as the joint

probability method (JPM). Its main advantage lies in its efficient use of source data; all values of the residual contribute to the final probability distribution function (PDF), even if they are not at or near high tide. However, there are two notable drawbacks to the JPM. First, converting the JPM output PDF to design risk is challenging, as it requires an understanding of flood duration. Second, and perhaps more critically, research has indicated that the residual and astronomical components are not statistically independent and are not sufficiently uncorrelated (Tawn and Vassie, 1989; Tawn, 1992), making the

convolution of these components questionable. The correlation becomes particularly evident during storm surges, as the timing of high tide is often shifted in relation to the underlying astronomical tide, which undermines the convolution. JPM is widely utilized (McInnes et al., 2013), though it has also faced criticism (Batstone et al., 2013) on the above grounds.

    The skew surge joint probability method, SSJPM, aims to overcome this shortcoming by using the difference between the maxima, rather than the difference between instantaneous height (Batstone et al. 2013). This new difference, referred to

as the "skew surge height," is claimed to have minimal correlation and is therefore considered a suitable candidate for characterizing surge statistics (Williams et al., 2016). However, there are only a limited number of skew extreme high tides, resulting in fewer output points compared to the Joint Probability Method (JPM). To accurately extrapolate the tail of the probability curve, an extreme value probability distribution, such as the generalized Pareto distribution (GPD), is needed. For a complete set of equations, the reader should refer to Batstone (2013). SSJPM was applied to tidal locations around the UK

but as Batstone reported: *"At approximately one-quarter of the UKNTGN sites, it became clear that the GPD fitted on the skew surge distribution was leading to a seemingly implausible representation of the most extreme sea levels."* The values were corrected by manual intervention using the AMAX method and also data values from nearby tidal stations.

    The above review highlights the need for a method of analysing tide gauge data that can consistently deliver design risk as a function of height, without requiring manual intervention. This paper discusses the development of such a method,

building on the AMAX approach but utilising more extremes each year, when applicable. The method initially identifies the highest sea level values recorded each tidal day rather than each year, leading to the acronym TMAX. These maxima are sorted and truncated, leaving a set of the most extreme, n values as inputs to the analysis. The paper discusses how a judicious choice of n reduces uncertainty in the resulting output; the flood return period as a function of height. The results of this TMAX



approach demonstrated good agreement with findings from a study commissioned by the UK Environment Agency 2011 which

used the SSJPM method at 35 identical coastal locations. However, unlike in the UK Environment Agency study, the TMAX method did not require manual intervention, running the same algorithm for every tidal location. The primary goal of this study is to evaluate the application of the TMAX technique, rather than to provide a conclusive re-evaluation of the UK flood defence heights.

Throughout this text, the term tide refers to the total sea level, while the astronomical tide refers to the deterministic

component caused by tidal forces generated by the earth-solar and lunar-earth orbits. The difference between the two is termed the residual, the storm surge, or the random noise component.

## 2 Background

W.E. Fuller (1914) claimed that, on a purely empirical basis, the size of floods increases proportionately to the

logarithm of time. Some 40 years later Gumbel, in his classic 1954 paper "Statistical Theory of Extreme Values and Some Practical Applications", theoretically justified Fuller's claim, provided the extreme values are derived from a stationary series, i.e. one whose mean did not drift uniformly with time, and have a probability distribution of an exponential type. This latter condition, later known as a Gumbel/Fisher-Tippett Type I distribution, applies but is not limited to exponential, normal, chi-squared, logistical and log-normal distributions. (See also Leadbetter 1983, Tawn 1988). Gumbel gave many practical

examples ranging from floods, radioactive decay, human life expectation, electrical capacitor breakdown, the strength of yarn, and the stock market share value. In Gumbel's original description (see his Eq. 2.17), a number, N, of observed peak values (generally annual), were ranked in ascending order with each ranked value i, being converted into the cumulative probability of a value not exceeding the ranked value, $F_i$ by the formula.

$$F_i = i / (N+1) \tag{1}$$

The probability F is related to the return period T (generally in years) by Gumbel's Eq. (2.8), i.e.

$$T = 1/(1 - F) \tag{2}$$


Gumbel showed that for a Type I distribution, the probability of the value being below a given value can be written as

$$F = \exp(-\exp(-y)) \tag{3}$$





The reduced variate, y exhibits a linear relationship with observed extreme value x, via a scale factor α and location factor, μ being given by

$$y = \alpha\,(x - \mu) \qquad (4)$$

Therefore, in a plot of observed extreme height x, versus reduced variate y, each extreme value falls approximately in a straight line. In Gumbel's time, each ranked value was physically plotted on probability paper, where the horizontal scale had been marked out according to equation (3), in terms of either return period or reduced variate or both, depending upon the manufacturer of the paper. A straight line was fitted to the points, and extrapolation of the line gave the probability of a value not exceeding a given value x, without explicitly requiring the calculation of the scale factor, α or location factor, μ.

Returning now to equation (1), Gringorten (1963) proposed a correction which improved the plotting accuracy and has been widely adopted as

$$F_i = (\,i - 0.44) / (\,N + 0.12\,) \qquad (5)$$

## 3.  TMAX Method

It has already been mentioned that there is a requirement for an accurate, consistent and reliable method of establishing the relationship between flood design risk and sea defence height. The proposed technique described here, named TMAX, is designed to meet this need and to be suitable for automation without needing manual intervention, and which uses the data efficiently. TMAX differs from the method described by Gumbel (and subsequently called AMAX) in three main ways. Firstly, a tidal day replaces the annual grouping used in AMAX (hence the name TMAX), secondly, only a subset of the total number

of peak values are used to fit the straight line in the probability plot, and finally a descending rank replaces the ascending order used by Gumbel. Further details are now described.

### 3.1 Tidal Day

Most but not all of the examples given by Gumbel use the annual maxima, which later became known as the AMAX method.

However, in his original paper, Gumbel also examines some extreme events without using annualized data, such as the breaking point of yarn, and the breakdown voltage of electrical capacitors. He states in his conclusion, "If the number of observed extremes N is not excessive, do not group the observations." Therefore, although extrema are generally grouped annually before applying Gumbel's method it is not necessarily the case. Annual grouping is thus replaced here with a time unit of one tidal day which, also known as a lunar day, is the time it takes for a specific site on Earth to rotate to the same point under the





moon and has a value of 1.034989 solar days. Changing the time unit from annual to tidal days considerably simplifies the identification and selection of tidal maxima which naturally occur in phase with the passing of each tidal day.

## 3.2 Descending Rank

Harris (1996) indicated there were some mathematical advantages in reversing the rank order as compared with Gumbel's original scheme, and this reversal is used here. This is particularly appropriate for this implementation since the most extreme
values are known but the least are not and would be much more difficult to reliably determine. However, the total number of observations, N, is precisely known, since it is the source duration in units of tidal days. We consider here F(x) to be the probability of the tide exceeding a height value x i.e. flood probability, rather than the opposite as in Gumbel's original formulation. Adding a prime to refer to the above equations (2 & 3) from Gumbel's original paper and substituting for Gumbel's descending rank, i' with an ascending rank, i, where i = N + 1 - i' and F '= 1 - F, hence equations (2 & 3) now become


$$T = 1 \, / \, F \qquad\qquad (6)$$
$$F = 1 - \exp( - \exp( - y )) \qquad\qquad (7)$$

where y is the reduced variate. Conversion from flood probability F to the reduced variate, y is carried out using the inverse of
equation (7), namely.

$$y = - \log_e( - \log_e( \, 1 - F \, )) \qquad\qquad (8)$$

Remembering that the reduced variate, y relates to flood height x via equation (4), this formula approximates to y = log(T)
since F is very small, confirming Fuller's original claim that maximum flood height varies with the logarithm of time. Since the tidal day has been adopted as the basic time unit, the flood return period is returned in tidal days; it can be converted to solar days by dividing by the length of a tidal day which is 1.034989 days. The reversal of rank order does not affect equation (1) nor Gringorten's correction equation (5), since it can be shown by making the above substitutions that the formulae remain the same and hence apply to both ascending and descending ranks. The Gringorten formula, rather than equation (1), was used
in all of the relevant calculations from hereon. The design risk D(x), i.e. the probability that a given value of x will be exceeded during a design life consisting of n durations (usually years), is given by.

$$D(x) = 1 - ( \, 1 - F(x) \, )^n \qquad\qquad (9)$$






### 3.3 Peak Selection

In practice, it is relatively easy to determine the largest peaks in a tide gauge record. As the data is sequentially scanned, a transition of sea-level above mean-sea-level (MSL) is identified, and the largest value is found and stored before the sea-level
scan returns below MSL. Continuing this scan generates a list of maxima. This list may include smaller peaks associated with noise and smaller semi-diurnal tides, but as only the most significant are ultimately used, their existence in the list is entirely irrelevant. Clearly, for a duration of tidal data of many decades, there will be many thousands of valid tidal-daily peak values. However, using all of the tidal-daily peak values on a probability plot is not advisable because of curvature in the distribution at less extreme values.  Figure 1 shows this curvature and how it influences a straight line fit. In harmonic tidal analysis,
shallow water coefficients are necessary to take account of non-linearity in the ocean response occurring at lower tides (Doodson and Warburg 1941).

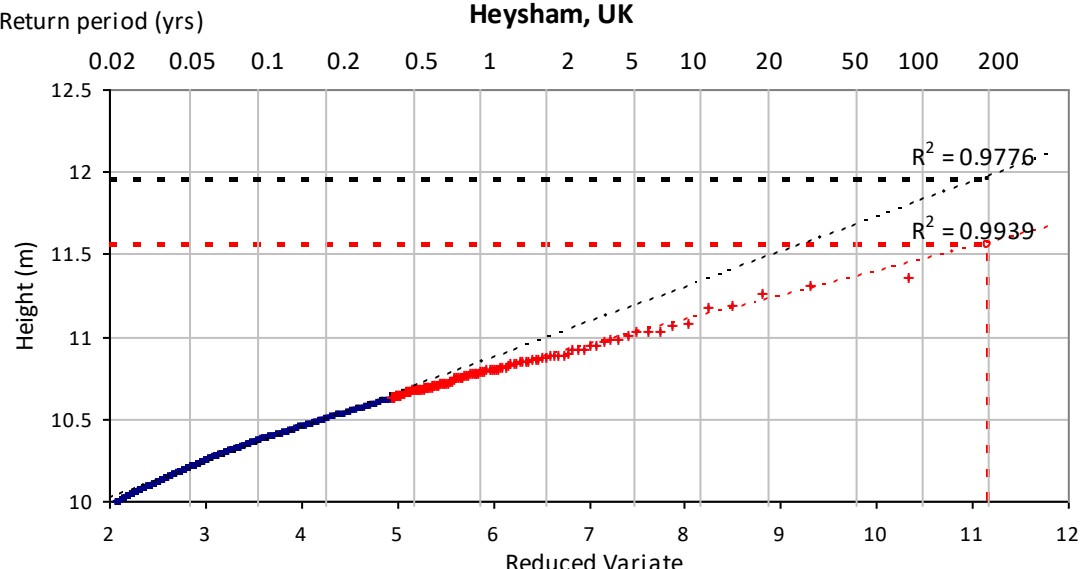

**Figure 1. Probability extreme tide plot showing its curvature in a 54-year record. Setting a threshold of 50 greatest extremes tides per year (2439 total), the LSQ intercept is 11.95m ± 0.02m R2= 0.9776 (black).  At 5 greatest extreme tides per year (242 points) the**
**straighter red section gives an improved LSQ intercept of 11.6m (red) ± 0.009m R2=0.9939. Location Heysham UK See 3.3 & 4.1 & Fig 3.**

Similarly, it is suggested that the curvature of the probability plot is probably related to shallow water effects during the neap of a spring-neap cycle.  Rather than attempting to fit the statistics of all high tides, we use a fit of only the n largest values, since these fall, more or less, in a straight line as shown in Fig. 1, where n is calculated from an average number per year, $n_A$
multiplied by the tide gauge record duration in years $n_y$, i.e. $n = n_A \times n_y$. The number of points used for the straight-line fit is much smaller than the number of high tides, N. Nevertheless, N must be used in equations (1) and (5) since it determines the scaling of the points plotted. The optimization of $n_A$ is discussed in Section 4.1. Note that this proposed method is quite





different from the Points Over Threshold (POT) method (Coles 2001), or R-Largest (Smith 1986). Here in the TMAX method like in the original Gumble AMAX method, ranking alone determines return period.

### 3.4 Straight Line Fit

The ordinary least squares (LSQ) method, also known as linear regression, was used to determine the line fit (Morrison 2021). Other methods exist including the method of maximum likelihood estimation (MME), method of moments (MOM), method of L-moments (MLM), method of probability-weighted moments (PWM), and the generalized least-squares methods GLSM/V. See Harris (1996), Coles (2000), Hong et al (2013), van Zyl & Schall (2012). Although it is stated that LSQ is less accurate than these other methods, LSQ uses simple closed-loop expressions to provide slope, intercept, and mean variance, providing a reasonable estimate. Consideration of using other fitting methods with TMAX is viewed as a potential refinement of the method for future research. With both AMAX and TMAX methods, the expected value of a predicted new point (i.e. a flood) is the mean y value of the intersection of the extrapolated regression line with the ordinate corresponding to the return period. Morrison gives expressions for the variance of the mean $\sigma_m$ and the variance of the new predicted mean $\sigma_p$, where n represents the number of data points, $x_m$, and $y_m$ are the mean x and y values respectively, as

$$\sigma_m = s^2_{y,x} \left( 1/n + (x_p - x_m)^2 / SS_{xx} \right) \quad (10)$$

$$\sigma_p = s^2_{y,x} \left( 1 + 1/n + (x_p - x_m)^2 / SS_{xx} \right) \quad (11)$$

where $\quad s^2_{y,x} = \left( 1/(n-2) \right) \Sigma(y_i - y_m)^2$

$\quad\quad\quad\quad SS_{xx} = \Sigma(x_i - x_m)^2$

The variance of the new predicted mean $\sigma_p$ is always greater than the variance of the mean $\sigma_m$ ; practically this is because a new point is subjected to variation in the process, whereas the variance in the intersection is subject to variation in the regression line parameters, the latter tending towards zero as the number of data points increases.

### 3.5 Missing Data

Frequently tide gauge data contains temporal gaps due to faulty instrumentation, or damage to the installation. Using the TMAX method, missing data can be taken into account by reducing the value of N to correspond to the number of tidal days of valid data. By contrast, it is not so straightforward in the AMAX method to handle missing data, since either synthetic data must be provided to fill the gaps, as was discussed by Gumbel, or whole years of data must be rejected if the annual data content is below a given threshold. This is a considerable advantage of the TMAX over the AMAX method.





## 4. Comparative Study

A comparison study was carried out of the AMAX, TMAX and SSJPM methods by repeating the study of Batstone 2013 at exactly the same 35 UK stations. The tide gauge data was downloaded and manually inspected, and isolated non-contiguous
data points and points showing clear evidence of gauge slips were removed. Most data before 1993 had been recorded at hourly intervals whereas subsequently, data was at 15-minute intervals. To obtain the longest data duration possible, both 15-minute and one-hourly data were used within a single file for many of the locations. Although there is a risk that the use of hourly data may result in missing or under-recording of the true peak value, most significant surges are at least one hour in duration, reducing this likelihood. The advantage of using much longer records should therefore easily outweigh the potential small
inaccuracy due to the change in sample rate within the file. In the first phase of analysis, the optimum value of $n_A$ was determined using the data processed above. In the second phase, the extreme sea levels corresponding to the return period were determined using the optimum value of $n_A$. These are now described and comparisons with the results from Batstone are indicated where appropriate.

### 4.1 Optimising the average number of peaks per year.

As discussed above, larger values of $n_A$ cause a higher proportion of extreme peaks under investigation to originate from smaller astronomical tides (neap tides) whose probability distribution may differ. Therefore, although larger sample values are generally associated with a decrease in variance, in this case the opposite may be true.

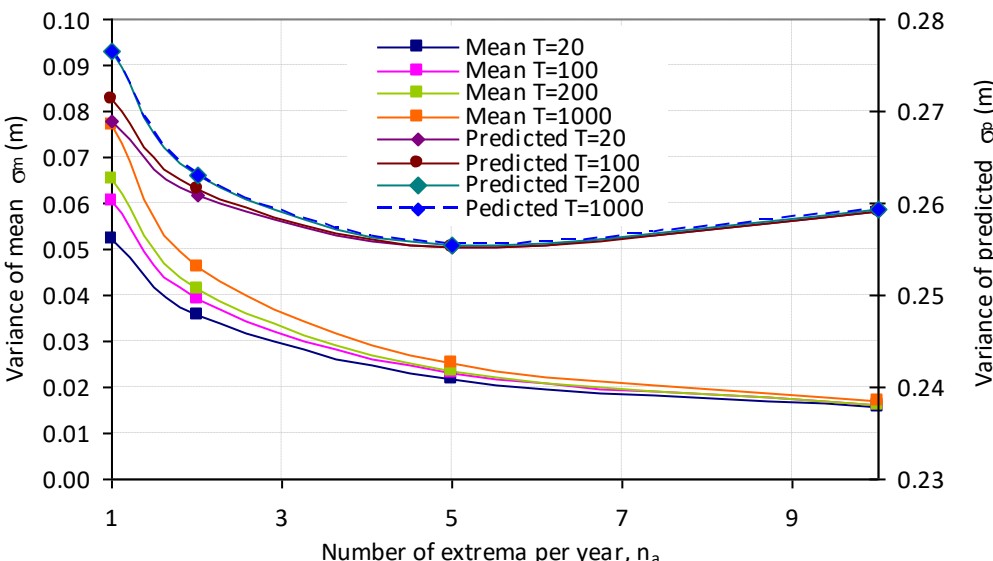

**Figure 2. The mean-variance $\sigma_m$ and of the predicted new point variance $\sigma_p$ plotted against the number of extrema per year $n_A$ for the TMAX method as practically determined from the tidal data.**





Figure 2 shows how in practice the variance of the mean, $\sigma_m$, and variance of a predicted new point, $\sigma_p$ varies with $n_A$, the average number of extremes selected per year, using the TMAX method. The values were determined from the tidal data using

equations (10) & (11). The variance of the mean, $\sigma_m$ decreases as $n_A$ increases; however, $\sigma_p$ shows a minimum with a value of $n_A$ of around five extremes per year. This concurs with the value of five per year found by Robson and Reed 1999 for extreme river flow studies and used by Smith 1986, although this may be coincidental. Nevertheless, since we obtain a minimum variance at the value of $n_A = 5$ this was adopted for all subsequent results. Table 1 illustrates the quality of fit of the regression lines for the two data durations and the AMAX and TMAX methods. The TMAX method, as compared to AMAX, gave a

higher quality of fit. The AMAX method showed a variance of the mean flood height value of around ±0.1 meters and a variance of a predicted new value of around ±0.4 meters, while the TMAX method gave lower values at around ±0.03 meters and ±0.24 meters respectively.

**Table 1: AMAX and TMAX methods. Quality of Fit of Regression Line**

| | Variance of Mean $\sigma_M$ (m) | | | | Variance of Predicted $\sigma_P$ (m) | | | |
|---|---|---|---|---|---|---|---|---|
| Return Period (yr) | 20 | 100 | 200 | 1000 | 20 | 100 | 200 | 1000 |
| AMAX to 2009 | 0.09 | 0.10 | 0.11 | 0.13 | 0.39 | 0.39 | 0.39 | 0.40 |
| TMAX to 2009 | 0.02 | 0.03 | 0.03 | 0.03 | 0.24 | 0.24 | 0.24 | 0.24 |
| AMAX to 2018 | 0.08 | 0.09 | 0.09 | 0.11 | 0.41 | 0.41 | 0.41 | 0.41 |
| TMAX to 2018 | 0.02 | 0.02 | 0.02 | 0.03 | 0.26 | 0.26 | 0.26 | 0.26 |

**4.2 Determination of Extreme Sea Levels**

Figure 3 shows a typical extreme tide probability plot for a single location, Heysham. The corresponding top four plotted points/heights are also listed in Table 2.

**Table 2. Tabulated data for the TMAX method at Heysham, UK.**

**Input duration span 19844 days, with valid data 17834 days i.e. 17232 tidal days.**

| Rank | Date | Height($H_0$) (m) | F(Ht>$H_0$) [*] per tidal day | Return Period [†] years |
|---|---|---|---|---|
| 1 | 01/02/2002 13:45:00 | 11.36482 | 3.2E-05 | 87.20 |
| 2 | 10/02/1997 13:15:00 | 11.30877 | 9.1E-05 | 31.30 |
| 3 | 30/03/2006 11:45:00 | 11.25951 | 1.5E-04 | 19.07 |
| 4 | 02/01/2018 23:30:00 | 11.18499 | 2.1E-04 | 13.72 |

[*] Calculated using Eq. 5 with N=17232 samples.

[†] Calculated using Eq. 6 with one tidal day=1.034989 solar days.





All of the results were derived directly from the tide gauge data and have not been re-processed in any way to enforce the
agreement with any other data set or with neighbouring local values. The analysis was carried out for the entire data set
consisting of 35 ports, once with the tidal records truncated to 1 Jan 2009 for comparison with the results of Batstone's analysis,
and once using the complete data set to 1 May 2018. Each data set was analysed using the AMAX and TMAX methods.  For
compatibility with Batstone, and in order to comply with the requirement for stationarity, all data was sea-level rise de-trended,
using the constant value adopted by Batstone of 2mm per year. The de-trending was arranged to apply a zero correction on 1
January 2008, to facilitate comparison with the results from Batstone who used the same date origin. The analysis used
Gringorten's correction throughout for compatibility with Batstone's research.

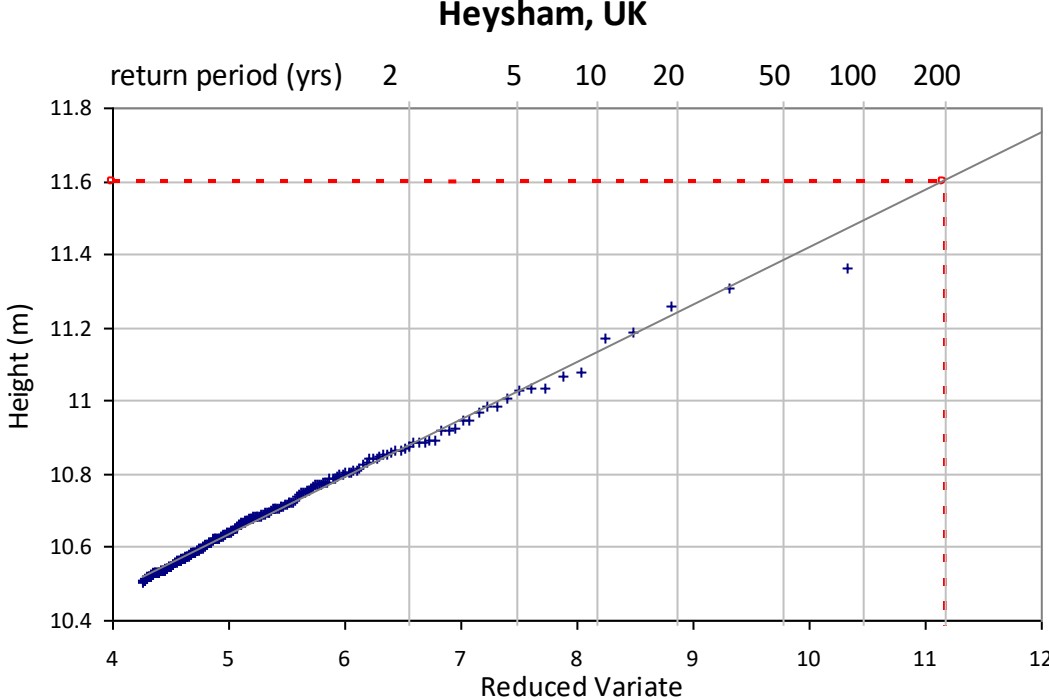

**Figure 3. Extreme value probability plot of 242 highest tides at Heysham over a 54-year duration, showing regression line. Height is
in meters relative to Admiralty Chart Datum (CD).  A 200-year flood return period gives 11.6m CD, 6.7m OD. (OD-CD Heysham =
4.9m). See Table 5.**

Table 3 compares the AMAX and TMAX results with those of Batstone. The mean differences for data to 2009 using either
method were generally in the centimetre region. In either case, the standard deviation was around 0.1m rising to approximately
0.24m for the 1000-year return level and was somewhat higher for the AMAX than for TMAX. Figure 4 shows the frequency



distribution of the differences between TMAX results derived from data to 2009 and data to 2018 across the 35 UK ports, showing a rise in level for each return period T.


**Table 3. AMAX and TMAX methods: Comparison with Batstone 2013.**

|  | Mean Difference (m) | | | | Standard Deviation (m) | | | |
|---|---|---|---|---|---|---|---|---|
| Return Period (yr) | 20 | 100 | 200 | 1000 | 20 | 100 | 200 | 1000 |
| AMAX to 2009 | 0.00 | 0.03 | 0.04 | 0.06 | 0.11 | 0.16 | 0.19 | 0.27 |
| TMAX to 2009 | -0.01 | -0.03 | -0.04 | -0.07 | 0.09 | 0.13 | 0.15 | 0.22 |
| AMAX- to 2018 | 0.02 | 0.06 | 0.07 | 0.09 | 0.10 | 0.14 | 0.16 | 0.22 |
| TMAX to 2018 | 0.02 | 0.01 | 0.00 | -0.02 | 0.09 | 0.12 | 0.14 | 0.20 |

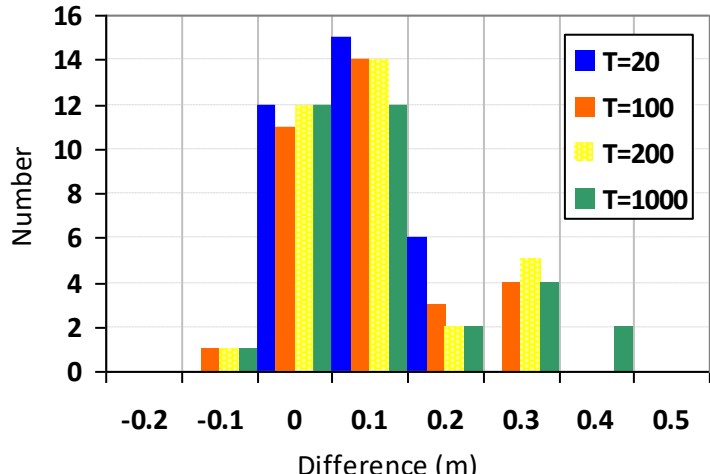

**Figure 4. Number of ports showing a given rise in the level of the return period between data to 2009 and data to 2018 as analysed across 35 UK ports.**

Table 4 indicates an average rate of rise over the 9 years of the extreme 20-year sea level of $3.4 \times 10^{-3}$ yr$^{-1}$ and for the 1000-year level of $6.1 \times 10^{-3}$ yr$^{-1}$. This compares with a rate of rise calculated for the whole dataset of $1.85 \times 10^{-3}$ yr$^{-1}$. When the increase is analysed by the paired-sample left-tailed t-test it gives P values of 0.005 to 0.006 indicating the increase in level is significant.



**Table 4. Rise in level averaged across 35 UK ports**
**between TMAX (2009) and TMAX (2018).**

| Return Period (yr) | 20 | 100 | 200 | 1000 |
|---|---|---|---|---|
| Mean Difference (m) | 0.031 | 0.041 | 0.044 | 0.055 |
| SD (m) | 0.066 | 0.088 | 0.096 | 0.121 |
| t score=M√N/SD | 2.706 | 2.678 | 2.653 | 2.623 |
| P (Significance=0.01) | 0.005 | 0.006 | 0.006 | 0.006 |


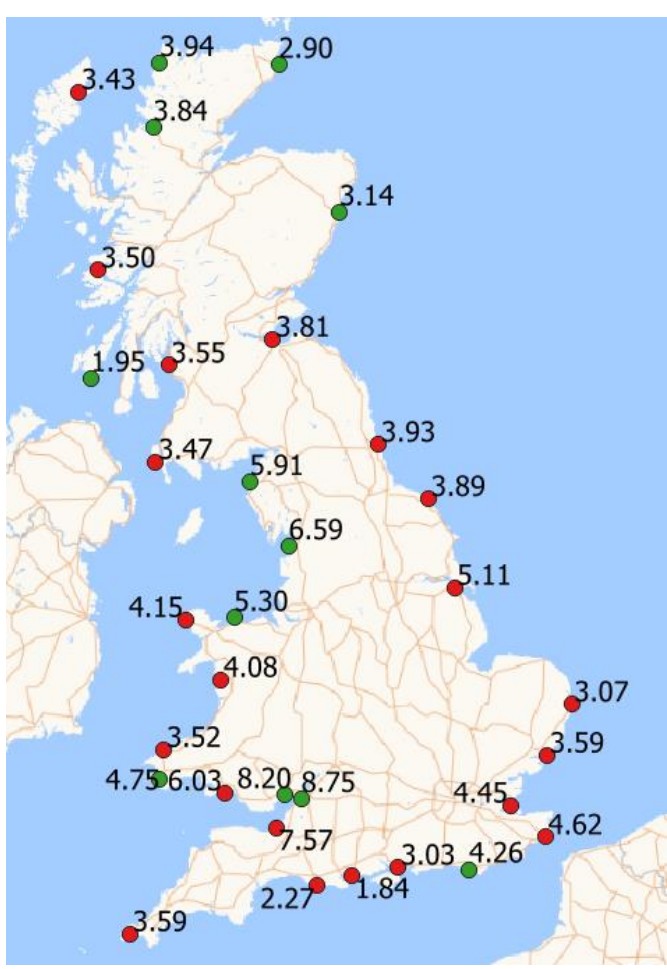

**Figure 5. Extreme Sea Level ODN (m) for a 100-year return**
**period at 35 selected locations in the UK. Colour indicates the**
**change between 2009 and 2018, red: increase, green: decrease.**




**Table 5: Extreme Sea-level (ESL) at UK coastal locations calculated from data to 2018 using TMAX method. The differences listed are relative to Batstone 2013 using data to 2009. Heights are in meters above Ordnance Datum Newlyn (ODN), relative to the mean sea level on 1 January 2008.**

| | Data | Return Period (yrs) | | | | Difference from Batstone (m) | | | |
|---|---|---|---|---|---|---|---|---|---|
| Location | Days | 20 | 100 | 200 | 1000 | D20 | D100 | D200 | D1000 |
| Aberdeen | 8798 | 2.99 | 3.14 | 3.20 | 3.35 | 0.02 | 0.03 | 0.03 | 0.06 |
| Avonmouth | 6227 | 8.53 | 8.75 | 8.85 | 9.08 | -0.14 | -0.23 | -0.26 | -0.35 |
| Barmouth | 8497 | 3.88 | 4.08 | 4.16 | 4.36 | -0.04 | -0.05 | -0.06 | -0.05 |
| Bournemouth | 6292 | 1.68 | 1.84 | 1.91 | 2.08 | 0.00 | 0.03 | 0.04 | 0.09 |
| Dover | 22578 | 4.35 | 4.62 | 4.74 | 5.01 | 0.11 | 0.14 | 0.17 | 0.21 |
| Felixstowe | 6384 | 3.27 | 3.59 | 3.72 | 4.03 | -0.06 | -0.13 | -0.18 | -0.34 |
| Fishguard | 13974 | 3.38 | 3.52 | 3.58 | 3.73 | 0.01 | 0.00 | 0.00 | 0.00 |
| Heysham | 17834 | 6.34 | 6.59 | 6.70 | 6.95 | -0.08 | -0.11 | -0.12 | -0.14 |
| Hinkley Point | 9077 | 7.41 | 7.57 | 7.64 | 7.81 | -0.10 | -0.17 | -0.20 | -0.28 |
| Holyhead | 15488 | 3.90 | 4.15 | 4.26 | 4.52 | 0.22 | 0.32 | 0.37 | 0.48 |
| Immingham | 20895 | 4.80 | 5.11 | 5.24 | 5.55 | 0.16 | 0.22 | 0.24 | 0.30 |
| Kinlochbervie | 7754 | 3.70 | 3.94 | 4.04 | 4.28 | 0.09 | 0.10 | 0.10 | 0.11 |
| Leith | 8453 | 3.67 | 3.81 | 3.87 | 4.02 | -0.02 | -0.07 | -0.10 | -0.18 |
| Lerwick | 12104 | 1.75 | 1.88 | 1.93 | 2.06 | 0.02 | 0.05 | 0.06 | 0.11 |
| Llandudno | 8046 | 5.10 | 5.30 | 5.38 | 5.58 | 0.01 | 0.01 | 0.00 | 0.00 |
| Lowestoft | 19538 | 2.69 | 3.07 | 3.23 | 3.60 | 0.04 | 0.00 | -0.04 | -0.18 |
| Milford Haven | 18158 | 4.57 | 4.75 | 4.83 | 5.02 | 0.09 | 0.08 | 0.08 | 0.07 |
| Millport | 12422 | 3.26 | 3.55 | 3.67 | 3.96 | 0.06 | 0.03 | 0.00 | -0.07 |
| Moray Firth | 3127 | 3.22 | 3.36 | 3.41 | 3.54 | 0.09 | 0.07 | 0.06 | 0.03 |
| Mumbles | 6121 | 5.85 | 6.03 | 6.11 | 6.30 | 0.02 | -0.02 | -0.04 | -0.09 |
| Newhaven | 11184 | 4.12 | 4.26 | 4.32 | 4.47 | -0.07 | -0.11 | -0.13 | -0.17 |
| Newlyn | 37171 | 3.43 | 3.59 | 3.66 | 3.81 | 0.10 | 0.13 | 0.15 | 0.18 |
| Newport | 8854 | 8.00 | 8.20 | 8.28 | 8.47 | 0.00 | -0.08 | -0.13 | -0.25 |
| North Shields | 18275 | 3.69 | 3.93 | 4.03 | 4.27 | 0.14 | 0.17 | 0.17 | 0.16 |
| Portpatrick | 17490 | 3.26 | 3.47 | 3.57 | 3.79 | 0.07 | 0.10 | 0.12 | 0.18 |
| Portsmouth | 8561 | 2.88 | 3.03 | 3.10 | 3.25 | 0.00 | -0.02 | -0.02 | -0.03 |
| Port Ellen | 6576 | 1.73 | 1.95 | 2.05 | 2.27 | -0.20 | -0.19 | -0.17 | -0.14 |
| Sheerness | 15999 | 4.21 | 4.45 | 4.56 | 4.80 | 0.08 | -0.02 | -0.08 | -0.25 |
| Stornoway | 12709 | 3.25 | 3.43 | 3.50 | 3.67 | 0.04 | 0.09 | 0.10 | 0.15 |
| Tobermory | 7890 | 3.32 | 3.50 | 3.57 | 3.75 | -0.15 | -0.24 | -0.30 | -0.43 |
| Ullapool | 16051 | 3.65 | 3.84 | 3.92 | 4.11 | 0.06 | 0.08 | 0.10 | 0.15 |
| Weymouth | 8739 | 2.13 | 2.27 | 2.32 | 2.45 | 0.08 | 0.07 | 0.06 | 0.05 |
| Whitby | 13192 | 3.70 | 3.89 | 3.97 | 4.16 | -0.08 | -0.13 | -0.17 | -0.25 |
| Wick | 18173 | 2.75 | 2.90 | 2.97 | 3.12 | 0.06 | 0.07 | 0.08 | 0.09 |
| Workington | 9017 | 5.66 | 5.91 | 6.02 | 6.28 | 0.10 | 0.10 | 0.11 | 0.13 |
| Mean | | | | | | 0.02 | 0.01 | 0.00 | -0.02 |
| Stdev | | | | | | 0.09 | 0.12 | 0.14 | 0.20 |


The values for sea level rate of rise do not include glacial isostatic adjustment (GIA) or geoid adjustments and are reasonably in line with reported values, (Wahl et al. 2013; Church and White 2011, IPCC 2021). Table 5 shows the heights using the TMAX method corresponding to the return periods of 20, 100, 200, and 1000 years for the 35 locations; it also lists the differences from Batstone's results. Figure 5 shows the ESL values for a 100-year return period deduced by the TMAX using

the data to 2018. The colours indicate whether the values for 2018 are higher (red) or lower (green) compared with the TMAX analysis to 2009. Investigation of the individual contributing tidal locations reveals four main outliers when comparing this 2018 TMAX study to Batstone's results for the 200-year return period (with heights shown in brackets) are; Tobermory (-0.3), Avonmouth (-0.26), Holyhead (0.37), Immingham (0.24). Attention is drawn to these locations as their ESLs may require revision.

**5. Conclusion**

When applied to UK tide gauge data, the described TMAX method generates reasonable estimates of extreme sea level return period and height, broadly in agreement with published values (UK Environment Agency 2011), with mean differences of the order of centimetres and standard deviation of the order of decimetres. However, the TMAX method is much simpler than the SSJPM used in EA2011. Unlike the SSJPM, TMAX does not use or require a) harmonic tidal analysis, b) the use of a

probability density function, or c) the fit of a generalized Pareto distribution (GPD), nor does it require manual intervention so is also amenable to automation. Compared to the well-known AMAX method, the TMAX method gives a significantly better internal fit and reduced variance. Missing data, quite common in tide gauge records, is also handled by TMAX in a much simpler, more efficient and elegant way, rather than with AMAX. The study indicates that the TMAX method is eminently suitable for coastal extreme sea level analysis and is at least as accurate as the AMAX and SSJPM methods. This innovative

approach can more effectively inform strategies for coastal management and resilience.

**Acknowledgments**

The author is grateful to the British Oceanographic Data Centre (BODC), who provided the tidal data used in this research via their "Sea Level Portal" at https://www.bodc.ac.uk. BODC do not accept any liability for the correctness and/or appropriate interpretation of the data or their suitability for any use.

**Competing Interests**

The author declares that he has no conflict of interest.



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
