# Peer review of "Evaluation of Extreme Sea-Levels and Flood Return Period using Tidal Day Maxima at Coastal Locations in the United Kingdom"

_EGUsphere, 2025_

## Referee Comment (RC3)

**Referee Report**

Evaluation of Extreme Sea-Levels and Flood Return Period using Tidal Day Maxima at Coastal Locations in the United Kingdom

**General comments**

The preprint proposes "TMAX," a workflow based on tidal-day maxima and linear regression on Gumbel reduced variates, positioned as an automated alternative to SSJPM and other established approaches. The topic sits squarely within the scope of *Ocean Science*, and the paper is motivated by an operational need for transparent and automatable estimation of return levels. In its current form, however, the scientific framing and validation are not yet sufficient for design-grade use: key assumptions (tail type, independence/declustering, stationarity) are not stress-tested; uncertainty is not fully quantified at the site level; and reproducibility is limited by brief methodological descriptions (peak selection, QC, and parameter tuning).

**Alignment with OS evaluation aspects (brief assessment).**

- Scientific significance. Addresses a relevant question and aims to deliver operational simplification; the novelty is primarily in workflow/automation rather than in new statistical theory. Substantial impact would require stronger validation and generalization beyond the UK testbed.
- Scientific quality. Core assumptions (Gumbel tail, independence of selected peaks, uniform de-trending) need diagnostics and comparisons (e.g., GEV/POT baselines). Results are compared against SSJPM but without per-site uncertainty bands or sensitivity analyses.
- Presentation quality. Generally clear narrative, but several essential methodological details are under-specified (peak selection, independence checks, QC). Figures would benefit from confidence bands and clearer indication of the fitted range.
- Reproducibility/traceability. The text should provide enough algorithmic detail (and QC rules) to enable replication; parameters selected by heuristics (e.g., number of peaks per year) should be justified or selected by diagnostics.

**Specific comments**

S1. Automation as primary justification vs. reliability for design. The manuscript motivates TMAX mainly by its potential for full automation as an alternative to SSJPM, which is presented as difficult to operationalize. If design decisions based on return levels may have severe consequences, methodological reliability should come first and automation second. Automation is valuable for broad regional screening, but for site-specific design one would typically prefer the best-supported method (SSJPM or POT/GEV with explicit diagnostics), even if harder to automate. Please either (i) provide stronger evidence that automated TMAX delivers design-grade performance (diagnostics, uncertainty, sensitivity, external validation) or (ii) clearly position TMAX as a rapid, approximate tool for comparative assessments rather than as a standalone basis for design.

- S2. Geographical scope and transferability beyond the UK. The paper frames TMAX as broadly automatable, but all evidence and tuning are limited to United Kingdom gauges. Please clarify whether the method is intended as UK-specific or globally transferable. If transferable, add an external validation or stress test at a few non-UK sites spanning different tidal regimes (micro/meso/macro; mixed/diurnal/semidiurnal), sampling cadences (15-min vs. hourly), and surge climatologies. Otherwise, state clearly that conclusions are restricted to the UK, and discuss what would be required to extend elsewhere (re-tuning of  $n_A$ , declustering windows, trend handling, tail-shape checks).
- S3. Assumption of Gumbel vs. GEV tail form (and relation to prior practice). By fitting a straight line on probability paper using the reduced variate, the analysis assumes a Gumbel (Type I) tail. The manuscript should test whether a GEV model with freely estimated shape  $\xi$  provides a better description (Gumbel, Fréchet, or Weibull). A POT (Pareto-Poisson/GPD) baseline is asymptotically consistent with GEV and would allow estimation of  $\xi$ . From a physical perspective, after de-trending, extreme still-water levels are plausibly bounded, which argues for exploring Weibull-type behavior ( $\xi < 0$ ) and could explain curvature in low-rank portions of the plots. The paper contrasts with work where heavy tails were mitigated via spatial averaging; here the tail difficulty is sidestepped by assuming  $\xi = 0$ . Please justify this choice with tail diagnostics (GEV and POT/GPD fits, QQ/PP/return-level plots) or relax the assumption.
- **S4.** Peak identification and independence of events. The peak-picking algorithm ("scan above MSL, store the largest value before returning below MSL") and the selection of the top  $n = n_A n_y$  peaks are described briefly, but independence (declustering) is not demonstrated. Please report extremal index estimates and/or runs-declustering sensitivity, and show ACF/dispersion checks to confirm that the retained peaks are approximately independent across sites.
- S5. Choice of sample vs. threshold modeling (link to r-largest/POT and threshold selection). Selecting the "top  $n_A$  per year" resembles r-largest approaches, and the choice of  $n_A$  plays a role analogous to threshold selection in POT. A single global variance minimum is unlikely to be optimal everywhere. Please provide sitewise sensitivity of return levels (with uncertainty) to  $n_A \in \{3, \ldots, 10\}$ , and outline diagnostics for automatic selection grounded in stability/predictive criteria (in line with established threshold-selection practice).
- **S6. Stationarity and de-trending.** All series are de-trended using a fixed 2 mm yr-1 with a specified origin date. This uniform correction may not match local relative sea-level change, and no diagnostic is provided for residual nonstationarity in extremes. Please add simple diagnostics (moving-window location/scale of daily maxima, change-point tests) or adopt a basic nonstationary model if indicated.
- S7. Uncertainty quantification for return levels. Formulas for  $\sigma_m$  and  $\sigma_p$  are given, but site-level return curves lack confidence or prediction bands. Please add per-site intervals for key return periods (e.g., 20, 100, 200 years) and discuss extrapolation uncertainty for longer periods.
- S8. Data curation and sampling resolution. Combining 15-min and hourly records is pragmatic, but potential peak under-sampling is asserted rather than quantified, and the QC procedure is described as manual. Please specify reproducible QC criteria and include a down-sampling experiment (e.g., degrade the 15-min periods to hourly and re-estimate return levels) to demonstrate that conclusions based on hourly data are consistent with those from 15-min data.

**S9.** Relationship to JPM/SSJPM and tide—surge dependence. TMAX sidesteps explicit modeling of conditional dependence between surge and tide. Please show whether selected peaks cluster near springs and whether dependence structures vary by site. If dependence persists after conditioning, consider incorporating it (e.g., via covariates in exceedance rate/parameters or via an extremal index correction).

---

## Author Comment (AC1)

**Calculation of extreme sea-level return period from past tidal maxima at selected United Kingdom coastal locations using modified Gumbel method.**

**S E Taylor**

**set@geomatix.net**

Date: 3rd May 2023

**Keywords: Tidal Flooding, Coastal Flooding, Sea-level Rise, Climate Change, Extreme Sea Level, ESL.**

REPLY to Anonymous Reviewer #1 in blue.

I sincerely appreciate all the time and effort spent by you, the "Anonymous Referee" in giving your valuable suggestions and correction to my paper. Although I may disagree with some of your points, it is clear that you have read the manuscript thoroughly, and I thank you for all your comments.

**General comments**

This study presents a new technique for extreme value analysis of coastal tide gauge records called the TMAX method. The TMAX method primarily improves upon the AMAX method of Gumbel & Lieblein (1954) by using a subset of the highest total water level values of each tidal day, rather than annual maxima. The author applies the TMAX method to the same 35 UK tide gauge records used by Batstone et al. (2013) for development of the skew surge joint probability method, and results are compared to extreme water level estimates derived from the SSJPM and the AMAX method of Gumbel & Lieblein (1954).

The author describes several valid advantages of this method, including: 1) the method is simpler than the SSJPM because it does not require harmonic analysis or fitting a probability distribution; and 2) using the highest water level of each tidal data, rather than annual maxima, enables more elegant treatment of incomplete time series.

It seems you accept the advantages mentioned in 1) and 2) above are valid. (I assume you mean "each tidal days' data" rather than "each tidal data".) However you then proceed to offer two main criticisms.

However, I disagree with one of the author's motivations for developing the TMAX method.

I am concerned regarding your disagreement with my motivation. My motivation was to attempt to develop a uniform, globally applicable method to provide estimates of flood return period from tide-gauge data, and which avoids manual intervention. Is it so misguided?

They write: "SSJPM is complex, often requiring manual intervention." It is true that the SSJPM is complex, but papers such as Batstone et al. (2013) and Baranes et al. (2020) clearly demonstrate that application of joint probability methods in regions where tides are large relative to surge (and thus large surge events may not be included in the highest recorded total water levels) provides more precise and stable return level estimates with a narrowed uncertainty range.

It seems you are suggesting here that some methods work better than others, depending on geographical features or surge-to-tidal-range ratio. That may well be, but nevertheless Batstone 2013 describes that in 25% of cases manual intervention was required and my attempt is to show that a simple re-incarnation of Gumbel's method with a small modification (TMAX) achieves, **on average,** similar if not better results across the board. That is not to say that splitting according to geographical regions or by surge to tidal-range ratios may not, in certain cases, achieve better results than my method.  I am happy to clarify this point in my text. It all depends on the aim and objective. It seems your objective is to produce the best possible result for a given location, irrespective of complexity of method and the level of manual intervention. This is a worthy and laudable aim.  However my aim is to produce a method of general applicability world-wide which is relavely easy to implement and which compares favourably in terms of accuracy with other methods now in use. I believe this is an equally laudable goal.

It would be more appropriate to apply the TMAX method in an area with smaller tides – or, perhaps, to apply the TMAX method to calculating skew surge statistics, and then convolving the TMAX probabilities with the tide probability distribution. In terms of the manual intervention, I interpreted that part of Batstone et al. (2013) as finding that manual intervention with the SSJPM was necessary in two distinct geographic regions (the Severn and James estuaries). To me, this points to a geographically linked phenomenon as a potential challenge (such as nonlinear tide-surge interaction or river influence), rather than the statistical method itself.

(I assume James is a typo for Thames.) Once more you are considering separating methods according to geographical regions.  It is well known that some regions of UK waters suffer from more from tidal surges than others e.g. the Southern North Sea and Severn Estuary, and some have a higher tidal range, e.g. Severn Estuary. However, it is not the purpose of my paper to assess the  effect of these factors upon the SSJPM study; perhaps this should be addressed by researchers in that field. However I do agree that a table indicating the manual intervention sites of Batstone and, the discrepancy with TMAX method on those sites, perhaps with a cartographic illustration, would be of interest to the reader of my paper and I propose to add this, to a redraft, if possible.

I interpret the primary conclusions of this paper as the TMAX method 1) giving "a significantly better internal fit and reduced variance" compared to the AMAX method, and 2) being "at least as accurate as the AMAX and SSJPM methods." In this paper's current form, I do not think that these conclusions are sufficiently supported.

It is clear from your above comments that I have not indicated the origin of my statistical claims.

- In general, there is little analysis provided of the results. In particular, it is important to discuss how geography, record length, and tidal range affect the results.

  I propose to redraft the conclusion section, justifying and supporting my conclusions. I had actually shown the record length in Table 3 but had not provided any discussion of it.

- I think "accuracy" is determined by consistency with the SSJPM, yet it seems that for 9 of the 35 locations, TMAX and SSJPM results are significantly different, even at the 20-year return period level. It is difficult to interpret this result without further discussion of factors such as record length and geography.

  I plan to discuss this point in the context of a statistical distribution examine if these values show any correlation with the factors you mention.

- It's a bit unclear how "internal fit" is compared, is it a comparison of the analyses that extend through 2009 to the ones that extend through 2018? If so...
  - On average, the TMAX-derived return water levels increase when the analysis is extended from 2009 to 2018 (Table 4). This is likely due to detrending the entire dataset using a constant assumed rate of RSLR that likely increased between 2009 and 2018. To actually assess something like stability (what I am interpreting "internal fit" to mean), it would make more sense to use something like a moving 365-day window to remove sea level rise and variability, then compare the results over the two time periods. Baranes et al. (2020) showed that application of a joint probability method significantly improved stability compared to fitting a probability distribution to total water level using a Monte Carlo validation method and by comparing statistical fits over two time periods. A similar sort of analysis could be used here to support the author's conclusion.
  - If I am interpreting "internal fit" correctly, the only comparison of the AMAX and TMAX internal fits are in Table 3, which only show mean values.

I will define/clarify/avoid the use of the words "internal fit" in a redraft.

Additional comments:

1. Fitting a constant sea level rise rate for the "AMAX to 2009" and "TMAX to 2009" analysis may facilitate comparison with the results in Batstone et al. (2013), but it will yield less robust statistics compared to using something like a moving 365-day mean, as rates of RSLR are variable, and inter-annual sea level variability may significantly affect the results. This should be discussed in the manuscript.

Yes, the constant sea level rise was used to facilitate comparison with Batstone. You twice mention the use of a 365 day moving averages for the determination of sea level rise. This method increases the "bandwidth" of the sea level rise value, so it can follow real variations, yet it also increases its susceptibility to random noise. There are other ways and means of

accounting for sea-level rise, I think it probably requires a paper to itself.  This aspect is interesting but probably out of the scope of the paper.

The use of a tidal day, rather than 1 year, is described as a novel approach. However, once the tidal days are reduced to a subset of the largest tidal days, I interpret this method as becoming equivalent to what Batstone et al. (2013) does in applying a peaks-over-threshold approach to skew surge (although this paper uses total water level). In other words, selecting a number of peak tidal day water levels that is equivalent to 5 times the record length is roughly equivalent to using the top 0.7%. This is thus equivalent to the innovation of the peaks-over-threshold approach, and the novelty here is overstated.

I agree that the TMAX approach described can be considered a special case of the POT approach. The additional special points here are a) the choice of a one tidal day "block time"; this assumes the peaks are sufficiently independant, b) it provides a more elegant approach in handling incomplete data sets and c) it utilises the choice of values down to a certain rank (suggested as 5) in the text which provides the correct balance between quality of fit and which avoids curvature of the line at lower amplitude values.  This can be seen as a strategy for selecting the threshold in the POT method. I also used the original Gumbel methods which relate the rank to frequency, using the Gringorten correction, and the least square fit rather than maximal likelihood.  I am not claiming the TMAX is innovative because it more complex and sophisticated.  However TMAX is unusual because it represents a return to Gumbel's original approach and to simplicity. The resulting very reasonable outcome by comparison with Batstone makes it all the more noteworthy.

In the specific comments below, I highlight several parts of the introduction with questionable descriptions of published studies, the methods description is confusing, or the results are inconsistent..

**Specific comments**

Lines 24-26: While tide gauge-based extreme value analysis is valuable for many reasons, the spatially varying nature flood return levels that you mention (lines 24-25), along with the fact that tide gauges are generally purposefully installed in wave-sheltered locations, make it somewhat rare that they are the only tool used to determine design elevations for coastal defence structures. Perhaps you could modify the text to describe alternative applications, such as for determining boundary conditions and/or validating numerical models used for coastal planning.

Good points. I will modify the text to include the discussion of surges, wave-heights and its dependence upon gauge position and design, and meteorological noise and residuals. I will also mention the use of tide-gauge data for model validation and determining boundary conditions.

Lines 29-31: This definition of HAT could use some clarifying. Do you mean that HAT assumes average conditions for the *meteorological component* of tidal height (or perhaps of total water level)? This might be clearer than calling the meteorological component of water level "noise."

HAT / noise / residual / wave height / meteorological component issues. Similar to previous.

Lines 42-43: I'm curious why this is being highlighted as a particular weakness of the JPM when none of the extreme value analysis methods you discuss in the introduction provide flood duration information.

The JPM, relies upon a convolution of the separate probability distributions for the deterministic predictable tide and for the residual components. However, the validity of this assumes the two distributions can be considered independent. Furthermore, the conversion of PDF output into design risk relies on assumptions regarding flood statistics. 👍

Lines 45-46: This is not quite right. The timing of the actual astronomical tide is shifted compared to the predicted astronomical tide (and the predicted tide is what's used to calculate the non-tidal residual).

Lines 49-51: It's not that the "difference" is uncorrelated; it's that skew surge is uncorrelated with measured high water (see Williams, 2016).

Lines 51-53: The SSJPM fitting a GPD to skew surges is not a reflection of there being fewer skew surge values compared to non-tidal residual values for a time series of the same length. Fitting the GPD (or any extreme value distribution), as opposed to an empirical distribution, has the advantage of providing probabilities for values that exceed the maximum observed value. In fact, the Revised Joint Probability Method (Tawn & Vassie 1989; Tawn, 1992) made this improvement by fitting a GEV to the non-tidal residuals (rather than an empirical distribution).

The above three comments i.e. Lines 45-53 refer to my rather brief description of the JPM and SSJPM which lacked accuracy. I will reword Lines 45-53. See also Technical Corrections below,

Lines 69-71: I would recommend defining these terms earlier in the introduction and using one consistent term for measured minus predicted water level. I recommend "residual," rather than storm surge (because the residual is often not storm surge) or random noise (because there are deterministic components of the residual).

HAT / noise / residual / wave height issues Similar to previous..

Lines 54-57: See general comments above about the Batstone et al. (2013) manual intervention

Already dealt with above

Lines 128-134: This paragraph is confusing in a couple of places:

- Lines 129-130: Do you mean the reverse? i.e. that the extreme values are more difficult to determine?
- Lines 81-83 (and Equation 1) show Gumbel ranking in ascending order, but you say "substituting for Gumbel's descending rank with an ascending rank"

I believe these statements are correct See Gumbel 1954 Eq 2.17 and 2.8 and Harris 1996. I should perhaps have worded it as " the extreme values are easier to index" so it is sensible to index the largest as number 1.  In Gumbels' original paper the values were indexed in ascending order with the highest value being indexed as m. Indexing the extreme values in descending order is simpler. Since the TMAX method described uses the top N extreme values, where N is shown to be about 5 per year, the index system can then populate the arrays using indices 1 to 5.

Lines 209-214: Why not fit a spline to the hourly data? Or you could show that it's not important to do this by comparing high waters over time periods with 15-minute data to time periods with hourly data.

A spline fit could help with uneven sampling rates, as is sometimes present in historical data. However I am somewhat concerned that a spline may, on occasion, create an incorrect peak value between two existing value.  I felt that on balance this is a little distracting from my general objective.

Table 3:

- Is the "mean difference" the mean of the difference between AMAX or TMAX and Batstone et al. (2013) across all 35 stations? Which is subtracted from which? And is the standard deviation the standard deviation of the difference?

I will clarify these points in the redrafted text.

- I would suggest showing these results on a map for each individual gauge – especially because Batstone et al. (2013) discusses individual sites where the GPD fit to the skew surge distribution was not physically plausible. Essentially, show the information in Table 5 on a map.

I do agree that a table indicating the manually intervention sites of Batstone and, the discrepancy with TMAX method on those sites, perhaps with a cartographic illustration,  would be of interest to the reader of my paper and I propose to add this, to a redraft.

- Why don't the means and standard deviations in Table 5 match the mean differences in the "TMAX to 2009" row of Table 3?

I will answer this point in my redraft.

Figure 4: This is difficult to interpret the way the bins are labelled and without geographic information. It should be shown on a map with exact values reported. The same should be done for AMAX (and AMAX should somehow be compared to TMAX) to support the conclusion that one method provides more stable estimates than the other (see also general comments above).

Results presented in Table 5: There are 9 sites that have differences greater than or equal to 10 cm at the 20-year return period level, compared to the SSJPM (Avonmouth, Dover, Hinkley Point,

Immingham, Newlyn, North Shields, Port Ellen, Tobermory, Workington). This seems like a relatively large difference, but it is difficult to interpret without information on geography, record length, or tidal range

I do not believe this number of outliers is statistically significant and will address this point in the redraft .

Conclusions: I interpret the primary conclusions as 1) the TMAX method giving "a significantly better internal fit and reduced variance" compared to the AMAX method, and 2) the TMAX method being "at least as accurate as the MAX and SSJPM methods."  These conclusions should be described in an expanded and quantitative discussion section that points to the specific results and/or analyses that support the conclusions. Topics such as how geography, tidal range, and record length impact the results should be discussed.

Actually record lengths are provided in Table 5. The conclusions will be written in a more rigorous way to justify the claimed statistics, along with a quantitative discussion of relevant factors.

**Technical corrections**

Lines 48-49: Perhaps revise to "... difference between the maxima of measured and predicted water level for each tidal cycle..."  Similar to Lines 45-53 above

Batstone et al. (2013) is sometimes referred to as "Batstone" and sometimes referred to as "Batstone 2013." Agree, consistency is needed here.

The equations used in the TMAX method should be more clearly and concisely stated.

They are stated in equations 1 to 9 spanning only two pages i.e. 70 lines. These pages also describe the Gumbel method of which the TMAX method is a relatively minor modification.  I fail to see how my description could be more concise.

Figure 5: I recommend not using red and green for colour blindness . Noted. Will amend.

S E Taylor

30 June 2025

---

## Author Comment (AC4)

**Reply to Remarks on Referee Report As posted on egusphere on 23 Oct 2025-10-23 by Roberto Minguez In response to**

"Evaluation of Extreme Sea-Levels and Flood Return Period using Tidal Day Maxima at Coastal Locations in the United Kingdom"

Stephen Taylor

Date: 3 November 2025

The referee states "the scientific framing and validation are not yet sufficient for design-grade use: key assumptions (tail type, independence/declustering, stationarity) are not stress-tested; uncertainty is not fully quantified at the site level; and reproducibility is limited by brief methodological descriptions (peak selection, QC, and parameter tuning). " and "Substantial impact would require stronger validation and generalization beyond the UK testbed".

Unfortunately the referee has missed the point of the paper. The study does not intend or pretend to be a world-wide exhaustive study of the proof of the described TMAX method in all situations; neither is it a request for international approval of the TMAX method. It is a case study comparing the results of extreme sea level calculations based upon either a method using the 75 year old Gumble's original method using Type 1 statistics (with a slight difference in selection method) to a single UK government study, using the SSJPM method.. The title indicates what it is, an evaluation of a single study.

It would be very difficult indeed to extend the study to other countries as the reviewer suggests because such detailed studies using the SSJPM method have not been carried out in most countries. Neither is the accompanying source data available over sufficiently long periods for the analysis by either TMAX or SSJPM for comparison. Therefore the broader type of study the reviewer suggests cannot be carried out in practice, without a major international effort.

It may make difficult, uncomfortable reading for some to accept that there has been little or no improvement in accuracy despite some 75 years of statistical tidal research, but that is the general conclusion of the study, and in this paper I back up this claim with sufficient data to warrant its publication.

The study was fully automated and required no manual intervention and this is presented, quite correctly I believe, as an advantage. However, the referee tries to turn this into a disadvantage stating "Automation is valuable for broad regional screening, but for site-specific design one would typically prefer the best-supported method (SSJPM or POT/GEV with explicit diagnostics), even if harder to automate." This broadside side-steps the main issue to be drawn from the paper.

The automated method described gave comparable or better results than the UK study, whose initial aim was to apply a general method, and yet which subsequently required manual intervention in 25% of the sites. This surely shows an advantage of the TMAX method described. It should not be falsely claimed as a disadvantage as the referee tries to make out. The advantage of the method is clear in this particular case.

It is unfortunate that it is "a particular case" and not a more general case, but that is because of the practical reasons explained above

I accept that, in order to prove the general applicability of the TMAX method, it could benefit from more testing and evaluation work, and that its readiness for general use is perhaps somewhat overstated in my paper. I have indicated in my response to a previous referee, and repeat it again here, that I am willing a willing to dilute these broader claims.

However, this should not detract from the main feature of the study.

The paper indicates that a 75 year old method (slightly modified) still holds up astonishingly well in comparison with one of the latest statistical flood risk methods, the SSJPM.

This surely is a valid and is a most important point of which readers of EGU Ocean readers should be made aware, and this point should not be hidden by obfuscation and statistical jargon.

Dr. S E Taylor.